# Comprehensive Identification of Glycosphingolipids in Human Plasma Using Hydrophilic Interaction Liquid Chromatography—Electrospray Ionization Mass Spectrometry

**DOI:** 10.3390/metabo11030140

**Published:** 2021-02-26

**Authors:** Karel Hořejší, Robert Jirásko, Michaela Chocholoušková, Denise Wolrab, David Kahoun, Michal Holčapek

**Affiliations:** 1Department of Analytical Chemistry, Faculty of Chemical Technology, University of Pardubice, Studentská 573, 532 10 Pardubice, Czech Republic; khorejsi@prf.jcu.cz (K.H.); Robert.Jirasko@upce.cz (R.J.); Michaela.Chocholouskova@upce.cz (M.C.); Denise.Wolrab@upce.cz (D.W.); 2Institute of Chemistry, Faculty of Science, University of South Bohemia in České Budějovice, Branišovská 1760, 370 05 České Budějovice, Czech Republic; dkahoun@prf.jcu.cz

**Keywords:** glycosphingolipids, lipidomics, mass spectrometry, hydrophilic interaction liquid chromatography, human plasma, lipid profile, sample preparation, fragmentation behavior

## Abstract

Glycosphingolipids (GSL) represent a highly heterogeneous class of lipids with many cellular functions, implicated in a wide spectrum of human diseases. Their isolation, detection, and comprehensive structural analysis is a challenging task due to the structural diversity of GSL molecules. In this work, GSL subclasses are isolated from human plasma using an optimized monophasic ethanol–water solvent system capable to recover a broad range of GSL species. Obtained deproteinized plasma is subsequently purified and concentrated by C18-based solid-phase extraction (SPE). The hydrophilic interaction liquid chromatography coupled to electrospray ionization linear ion trap tandem mass spectrometry (HILIC-ESI-LIT-MS/MS) is used for GSL analysis in the human plasma extract. Our results provide an in-depth profiling and structural characterization of glycosphingolipid and some phospholipid subclasses identified in the human plasma based on their retention times and the interpretation of tandem mass spectra. The structural composition of particular lipid species is readily characterized based on the detailed interpretation of mass spectrometry (MS) and tandem mass spectrometry (MS/MS) spectra and further confirmed by specific fragmentation behavior following predictable patterns, which yields to the unambiguous identification of 154 GSL species within 7 lipid subclasses and 77 phospholipids representing the highest number of GSL species ever reported in the human plasma. The developed HILIC-ESI-MS/MS method can be used for further clinical and biological research of GSL in the human blood or other biological samples.

## 1. Introduction

Glycosphingolipids (GSL) are ubiquitous membrane components [1] predominantly found in the outer leaflet of the cell plasma membrane [2,3,4] of virtually all eukaryotic species [5] and some bacteria [6], where they are included in lipid rafts [1,5] together with cholesterol clusters, which serves as binding sites or receptors [4]. Furthermore, they have been found in membranes of intracellular organelles [3], and they also circulate in serum either in free or protein-bounded form [7]. These amphipathic compounds represent one of the major and most structurally heterogeneous subclass of sphingolipids [8,9] comprised of a hydrophilic head group with at least one monosaccharide residue attached via glycosidic linkage to a hydrophobic ceramide moiety [1,2,4,10]. GSL are immensely complex due to multiple variations in composition, binding positions, branching [2,11] and modifications (e.g., sulfation, sialylation) of a saccharide core [8] as well as in ceramide backbone (e.g., hydroxylation) [12]. GSL are commonly categorized into neutral GSL and acidic GSL according to the nature of a saccharide head group [4]. Neutral GSL consist of two subgroups: monoglycosylceramides (cerebrosides) and oligoglycosylceramides (globosides) [1]. Acidic GSL are subdivided into sialosylglycosylceramides (gangliosides) and sulfoglycosylceramides (sulfatides) [1,4,11].

GSL are not only cellular membrane components essential for the structural integrity of cells [13], but they also serve as important bioactive compounds with a variety of cellular functions including cell-cell and cell-matrix interactions, cell-cell adhesion, cell migration, growth, proliferation, differentiation, intracellular and extracellular signaling, and apoptosis [1,9,13,14,15,16]. More interestingly, GSL have been shown to undergo specific changes in disease-specific manner [17]. In the plasma membrane, GSL play a role in infectious diseases [5,10], where they act as cellular receptors and co-receptors for viruses, bacteria, and microbial toxins [11]. The accumulation of one or multiple GSL [4,14] negatively affecting cells and tissues of certain organ systems [5,16] is characteristic for Fabry or Gaucher disease, collectively called as storage diseases [7,16]. Moreover, alterations in GSL structure and their levels have been attributed to a wide spectrum of other disorders including folding diseases (e.g., Alzheimer and prion diseases) [11,15,17], skin disorders [17] diabetes (i.e., downregulation of insulin receptor) [13], cardiovascular disease [17] (e.g., atherosclerosis) [18], cystic fibrosis [15], and notably in various types of cancer [13,17]. In many cancer cells, there is a large number of proteins, especially enzymes, which alters the cell metabolism [13,19]. Similarly, GSL contribute to the cancer progression as they also have a key role in many pathways of cell metabolism [20], making them potentially useful biomarker molecules [11].

Since changes of GSL concentrations and modifications in the structure of GSL molecules have been implicated in various diseases, specific approaches for the isolation, detection, and structural characterization of as many GSL species as possible are highly important [2,9,21]. Biofluids, namely blood, are well suited for the biomarker search as (1) samples are easily drawn by a less invasive procedure, and (2) a proportion of GSL species can end up in bloodstream due to their segregation from surface of diseased tissues or cells during pathological states [22]. Approaches using GSL and their metabolites might be beneficial for finding novel potential diagnostic biomarkers and use them as diagnostic tools and targets for the cancer treatment in immunotherapy [4,13,19,23]. In addition, it may help with the understanding of pathogenesis, etiology, and major clinical manifestation of associated diseases [4,14], and may be also valuable for monitoring the progress of the therapy [14]. Lipidomic analysis of biological samples is routinely processed by either the direct analysis without chromatographic separation (i.e., shotgun lipidomics) or liquid chromatography coupled to mass spectrometry (LC/MS) [18,24,25]. In past decades, various lipidomics methodologies based on tandem mass spectrometry (MS/MS or MS^n^) [25] have been developed to enable a comprehensive analysis of a large number of sphingolipids [14,26]. Advancements in the lipid analysis by MS make this method dominant for a wide range of lipidomic applications [25,27], including the quantitative and structural analysis of GSL accomplished by MS/MS after the separation by either thin-layer chromatography (TLC) or high-performance liquid chromatography (HPLC) [5,18].

In case of biofluids, the GSL are usually less abundant in comparison with other lipid subclasses [28] and suffer from low ionization efficiency (especially the heavily glycosylated GSL) partially caused by decreased hydrophobicity and volatility [29,30]. Additionally, the amounts of individual GSL species differ by many orders of magnitude [31], particularly disease-specific GSL are often extremely low abundant, which requires more sensitive strategies to detect them including powerful extraction and separation methods before the MS analysis [28]. Nonetheless, progress has recently been made in the GSL analysis with better sensitivity due to a step forward in MS [32]. To address this issue, liquid chromatography coupled to electrospray ionization tandem mass spectrometry (LC-ESI-MS/MS) employing triple quadrupole (QqQ) or tandem quadrupole-linear ion trap (QTrap) analyzers respectively, or for higher mass accuracy, quadrupole-time-of-flight (Q-TOF) or Fourier transform (FT) instruments [26,33] is the major and the most frequently applied analytical method capable to provide an in-depth analysis of GSL in different biological samples with sufficient sensitivity [9,11] on both qualitative and quantitative levels [18]. Matrix-assisted laser desorption ionization (MALDI) has been reported as an optional technique [34]. Even though the GSL represent potential biomarkers, their analysis can frequently prove challenging, even when a suitable analytical method is available, as these molecules are amphipathic and immensely complex. The major challenge arises from the sample preparation. Notably, the extraction and purification of GSL from biological matrices have a crucial role for the successful lipid analysis [35].

In biological samples, GSL co-exist with major lipid subclasses (mainly phospholipids) and biopolymers (proteins and nucleic acids), and are therefore less abundant [36,37], as they compete for the ionization [38]. Thus, removal of undesired matrix effects together with co-present salts during the sample preparation procedure using an appropriate liquid-liquid extraction (LLE) and/or solid-phase extraction (SPE) approaches is highly desirable [18,24,37]. The commonly used procedure to extract lipids from biological samples is carried out through biphasic solvent extraction protocols described by Folch, Blight and Dyer, or butanol-methanol (BUME) methods [28,35]. However, these solvent systems are ineffective to extract more hydrophilic GSL as they require a higher aqueous proportion, therefore, the lipid solubility should also be taken into account [28]. In general, no extraction protocol is capable to simultaneously extract all GSL species with uniformly high yield due to the structural diversity. Nevertheless, Lydic et al. [37] came up with monophasic chloroform-methanol-water (1:2:0.74, *v*/*v*/*v*) solvent system facilitating the simultaneous analysis of highly polar and nonpolar lipids using the shotgun method. This approach provided lipid profiles comparable to those obtained by well-established biphasic extraction methods, and thus enabled detailed lipidome analyses of broad range of GSL species without necessity for multi-steps extraction methods, where the partial redistribution between two phases is commonly inevitable [37,39].

Additionally, a few methods are effectual in the separation of particular GSL subclasses [18,28,31]. These include a silicic acid chromatography, where phospholipids can be retained on a column while modified (acetylated) GSL are eluted out. The fraction with GSL may be further separated into neutral and acidic GSL using weak anion-exchange (WAX) column, and the resulting fractions are then purified by C18- or C8-based SPE [40]. Furthermore, aminopropyl and silica gel SPE cartridges can also be used to fractionate crude lipid extracts. Aminopropyl cartridge has been used to isolate cerebrosides and globoside [41], while silica gel cartridge may be used to enrich sulfatides from extracts [42]. Less common approach using the elimination of major lipids, such as phospholipids and acylglycerols through alkaline hydrolysis enables the analysis of low-abundant GSL species, as they are to some extent resistant to mild alkaline conditions [9,43].

In addition, chemical derivatization strategies are frequently used to enhance the sensitivity for more accurate quantification [28]. In respect of these facts, the utilization of monophasic water-soluble organic solvent systems seems to be the most favorable approach for the effective extraction of a wide range of GSL subclasses since some lipid species can be lost in the solvent system interface and/or their amounts may be decreased due to multi-steps extraction or redistribution between two phases when using traditional biphasic lipid extraction. It has been proved that the recovery of GSL is improved by the addition of water during the extraction step [11,25]. Furthermore, the mild acidification of mobile phase reduces the adduct formation in ionization source with counter ions like sodium, potassium, ammonium, and others [11]. The comprehensive analysis of GSL extracted from body fluids still poses a significant challenge [44] despite a considerable progress and development in methodologies for isolation and analysis of GSL [11,14]. Last but not least, a lack of commercially available internal standards (IS), particularly for more complex GSL, represents the limitation that obstruct the quantitation of GSL as well [28]. Consequently, the comprehensive analysis of GSL is not yet a routine practice to date [9,11].

In this study, a monophasic ethanol-water solvent system, as a safer alternative to chloroform, followed by C18-based SPE is applied for the isolation, purification, and concentration of GSL from human plasma. The subsequent structural characterization of GSL is performed by HILIC-ESI-LIT-MS/MS. The fragmentation behavior of individual GSL species in MS/MS and MS^n^ mode is studied in detail to obtain detailed structural information on the saccharide sequence composition and the constitution of ceramide part of GSL molecules necessary for the accurate lipid profiling. The use of traditional linear ion trap technology is beneficial for this lipidomic analysis due to its major advantages as high-ion storage capacity, high scan rate, and low costs. As a consequence, high-sensitivity full scan mass spectra are provided, which simultaneously improves MS/MS and MS^n^ possibilities important for the structural confirmation of individual lipid species.

## 2. Results and Discussion

### 2.1. Nomenclature of Investigated GSL

The structural classification of GSL is based on the saccharide sequence attached to a ceramide moiety (Figure 1). The first monosaccharide unit linked to the ceramide (Cer) is β-linked galactose (Gal) or glucose (Glc) resulting in galactosylceramide (GalCer, i.e., HexCer) or glucosylceramide (GlcCer, i.e., HexCer), respectively. The saccharide core may be extended by additional saccharide units to form more complex GSL, such as lactosylceramide (LacCer, i.e., Hex_2_Cer), globotriaosylceramide (Gal-Gal-GlcCer, i.e., Gb_3_), and globotetraosylceramide (GalNAc-Gal-Gal-GlcCer, i.e., Gb_4_). MS experiments do not provide an evidence to differentiate Glc and Gal, therefore we use only the symbol Hex for hexose in mass spectra annotation in this work. The numbers after Cer (e.g., Cer 18:1/16:0) report the total number of carbon atoms and double bonds (CN:DB) in the sphingosine base (e.g., 18:1) and N-linked fatty acyl (e.g., 16:0) of the ceramide moiety. In case there is an additional hydroxyl group in the ceramide part without any specification of its position, the OH is placed behind the ceramide DB number (e.g., 16:0 OH). GalCer and Gal-GlcCer analogues with a sulfate group attached to C3-hydroxyl of galactose are called sulfatides (i.e., monohexosylsulfatides, SHexCer and dihexosylsulfatides, SHex_2_Cer). Sialylated GSL, such as monosialodihexosylgangliosides (GM_3_), are called gangliosides, and their shorthand notation has already been thoroughly described in our previous work [45].

### 2.2. Optimization of Sample Preparation

There is no single extraction protocol able to extract all GSL species in a high yield. Here we focused on the utilization of modified chloroform-free monophasic solvent system as a suitable alternative for the isolation of GSL in the human plasma. For the detection of trace GSL species, we had to implement the purification of plasma extracts from proteins, non-polar lipids, salts, and other contaminants commonly present in plasma, because they may affect the sensitivity of GSL analysis. For this purpose, the effects of SPE columns, deproteinization solvents, and the effect of methanol abundance in the loading step were investigated. The optimization of sample preparation was performed based on the comparison of intensities of most abundant species within particular GSL subclasses.

#### 2.2.1. Comparison of SPE Columns

In previous works [45,46], HILIC-ESI-MS/MS analytical method for the determination of gangliosides and other polar lipids was developed. This method used 200 mg of Sep-Pak tC18 cartridge (Waters, Milford, MA, USA). In this work, we conducted a SPE comparative study with the objective to extract neutral and acidic GSL subclasses in a sufficient yield for lipid profiling and remove possible interferences. As a part of our pre-screening SPE column selection, we used different types of SPE cartridges with reversed-phase, normal-phase, polymeric-phase sorbent, and SPE with ZrO_2_-based sorbent. A more detailed specification of used SPE cartridges is listed in Appendix A. In our pre-screening measurements, we found that polymeric-based SPE (namely Strata SDB-L, Strata X, and Oasis HLB) provide lower intensities compared to reversed-phase based SPE. Moreover, we observed a considerable loss of targeted analytes during the loading step, when using polymeric-based SPE columns. Furthermore, the normal-phase based SPE (i.e., diol) provided the worst results from all tested SPE cartridges. ZrO_2_-based SPE cartridges (namely Phree and HybridSPE) were also tested. These are relatively selective in the removal of phospholipids, but we recorded a substantial decrease in peak intensities for investigated GSL subclasses. Consequently, we focused on a more detailed selection of C18-based SPE columns, which proved to be the best option for isolation, purification, and concentration of GSL from human plasma.

In total, seven C18-based and one C8-based SPE cartridges from different vendors, including the Sep-Pak tC18 cartridge, were compared and examined in more detail. SPE cartridges contained the same amount of sorbent (500 mg; except for 200 mg in case of Sep-Pak tC18). The SPE procedure was performed using the same extraction protocol (see Materials and Methods). Three replicates, each with three consecutive injections, were used for all types of C18-based SPE cartridges (Figure 2). Previously used tC18 Sep-Pak cartridge for gangliosides did not prove to be the best option for the isolation of neutral GSL, probably caused by smaller amount of sorbent in comparison to other cartridges. The highest intensities for all investigated neutral GSL subclasses are achieved with the DSC-C8 cartridges. However, error bars indicate a high variability in obtained results, which is unsatisfactory. The optimal results with satisfactory error bars for all investigated neutral GSL subclasses are realized with Spe-ed C18/18 SPE cartridges, which were selected for the next experiments.

#### 2.2.2. Comparison of Deproteinization Solvents

The selection of water-soluble organic solvent for a single-step deproteinization was optimized using methanol (MeOH), ethanol (EtOH), acetonitrile (ACN), and acetone (ACE). Three replicates for each solvent with three consecutive injections were performed (Figure 3A). ACN deproteinization is not an effective method for GSL extraction due to observed significant decrease in intensities of lipid species, which is in agreement with previous works [18,25]. Moreover, the ACN protein precipitation is probably not sufficient to precipitate all proteins in sample, which was noticed during the reconstitution step, where the addition of reconstitution solvent (MeOH) to dried sample formed a slightly cloudy solution. The use of ACE provides better results, but still not satisfactory due to large error bars. The alcohol-based protein precipitation was proved to be a superior deproteinization approach in the sample preparation procedure for LC/MS based GSL analysis. In our study, there was no significant difference between MeOH and EtOH, which confirms previous conclusions [25]. Absolute intensities and also error bars are slightly better for ethanol, therefore ethanol is selected as suitable protein denaturing reagent in this simple and quick single-step deproteinization approach. In addition, our outcomes support the fact that alcohol-based protein removal is traditionally applied in case of Folch and Bligh-Dyer methods, as well as for methyl-tert-butyl ether (MTBE)-MeOH method.

#### 2.2.3. Effect of MeOH Abundance in Loading Step

The final step in the sample preparation was the optimization of MeOH content in the loading step. Six levels of MeOH percentage (0, 1, 3, 5, 10, and 20) in the loading step were evaluated to obtain the highest possible response for each investigated neutral GSL subclass as well as to support the solubility of individual lipid species due to their amphipathic nature (Figure 3B). The addition of MeOH in the loading step causes a small decrease in the response of inspected GSL subclasses. Additionally, the MeOH is used as an eluent, which could cause undesired elution of a small amount of lipid species already in the loading step. Therefore, only pure water is used in the loading step.

In summary, the optimum conditions for the sample preparation were as follows: ethanol deproteinization with the subsequent SPE procedure using Spe-ed C18/18 SPE cartridge and water as the sample solvent in the loading step.

### 2.3. Optimization of Collision Energy

The optimization of collision energy of MS/MS transitions of neutral GSL was accomplished using the commercially available TLC neutral GSL mixture (Matreya). The collision energy was tested as normalized collision energy (NCE) on the scale from 10 to 100% with an increment of 10% based on GSL species with 34:1 ceramide backbone within four investigated neutral GSL subclasses. A 100 times diluted solution of commercial standard was directly infused into the Velos Pro mass spectrometer, and the optimal normalized collision energies for each transition were obtained by plotting relative intensities of individual precursor and product ions at given value of normalized collision energy, as illustrated in Appendix A. Taking into account the measured values and curves depicted in Appendix A, we have deduced that the optimal NCE for all involved GSL subclasses was achieved at 40% NCE. This fact is indicated by an extensive fragmentation of individual singly charged protonated molecule [M+H]^+^ producing characteristic product ions that are consistent with a sequential cleavage of respective saccharide units from the hydrophobic ceramide backbone.

### 2.4. Separation of Lipid Subclasses in HILIC Mode

The HILIC based LC/MS method and sample preparation procedure used in this study was based on conditions reported earlier [45] with modifications described in Experimental section. Figure 4A represents LC/MS reconstructed ion current (RIC) chromatograms obtained for the analysis of human plasma extract, where 4 neutral GSL subclasses (HexCer, Hex_2_Cer, Gb_3_, and Gb_4_), 3 acidic GSL subclasses (SHexCer, SHex_2_Cer, and GM_3_), and 4 phospholipid subclasses (phosphatidylinositols, PI; lysophosphatidylinositols, LPI; phosphatidylethanolamines, PE; lysophosphatidylethanolamines, LPE; and their ether-linked species called plasmalogens) were detected together with ceramides (Cer) and fatty acids (FA). We also detected other lipid subclasses that are eluted after 10 min run, for instance, phosphatidylcholines (PC; 12.3 min), lysophosphatidylcholines (LPC; 16.6 min), and sphingomyelins (SM; 15.2 min). These later eluting lipid subclasses are not investigated, because this work is focused mainly on GSL. The sample preparation procedure is not capable to remove all phospholipid interferences, hence we also investigate PI and LPI classes with the similar structural features as for GSL, because they also contain the saccharide part (i.e., inositol). In addition, they also co-elute with important GSL subclasses (namely HexCer and GM_3_), therefore PI and LPI may affect an ionization process of these inspected GSL subclasses.

Investigated lipid subclasses, such as HexCer, Hex_2_Cer, Gb_3_, and Gb_4_, provide singly charged [M+H]^+^ ions in the positive polarity mode, while SHexCer, SHex_2_Cer, GM_3_, PI, and LPI are better ionized in the negative ion mode as singly charged [M-H]^−^ ions. We should also note that all lipid subclasses detected in the negative polarity mode were observed in the positive polarity mode as well. In addition, the ammonium adduct [M+NH_4_]^+^ and/or sodium adduct [M+Na]^+^ were observed for some lipid species, mainly those most abundant. The chromatograms depict the specific chromatographic retention behavior of individual lipid subclasses following the regular patterns related to the type of polar head group and the number of attached saccharide units. The HILIC mode also enables the partial separation of individual lipid species within particular lipid subclass, so that retention slightly decreases with the increasing length of fatty acyls, as illustrated in Appendix A and previously demonstrated for ganglioside subclass in our recent work [45]. The retention of neutral GSL (red line) specifically increases with an increasing number of monosaccharide units. The increase is about 0.5–1 min for one monosaccharide unit, except for Gb_4_ where the retention is strongly increased by about 4 min due to N-acylation of the fourth monosaccharide residue resulting in more polar molecule. The retention of acidic GSL and phospholipids increases as the hydrophilicity of the molecule is increasing. Furthermore, the partial separation of GlcCer and GalCer was observed, when analyzing the neat solution of IS (data not shown), therefore we assume the separation is probably also achieved in plasma samples. However, we cannot clearly distinguish GlcCer and GalCer species in plasma with respect to identical fragments provided in MS/MS experiments.

Interestingly, Gb_4_ subclass is separated into 3 partially resolved peaks (see Figure 4A) corresponding to lipid species with long N-linked fatty acyl chains (earlier eluting peaks of Gb_4_) and short N-linked fatty acyl chains (later eluting peaks of Gb_4_). Nevertheless, MS and MS/MS spectra do not provide conclusive data for the interpretation why Gb_4_ species with long and short N-linked fatty acyl chains are distributed into two peaks. This may be probably caused by the presence of isomeric form of Gb_4_ denoted as iGb_4_ (i.e., isoglobotetraosylceramide), but no suitable standard is available to differentiate these two subclasses. Figure 4B shows that *sn*-1 and *sn*-2 regioisomers were detected for both lysophospholipid subclasses (i.e., 1-LPI/2-LPI and 1-LPE/2-LPE). The *sn*-1 regioisomers had higher retention times than their *sn*-2 counterparts, as described in the previous work [47]. The elution order presented in this study is in agreement with numerous previous works [7,9,14,15,45,46], therefore the characteristic retention behavior based on above mentioned rules is used as the supporting information for the lipid identification.

### 2.5. LC-ESI-MS/MS Characterization of Lipids

The structural characterization of lipids is based on retention times and *m/z* measurements of precursor and products ions using the characteristic fragmentation pathways in positive (neutral GSL) and negative (acidic GSL and phospholipids) ESI mode. The fragmentation pathways of most abundant neutral GSL have been characterized in previous works [2,8,9,10,14,15,21], but comprehensive MS/MS description of all detected GSL species in human plasma, which could speed up the identification of lipids, is not yet available. Therefore, we provide an extensive characterization of detected neutral and acidic GSL and also detected phospholipids. The used fragmentation nomenclature is consistent with the standard designations according to previous works [48,49,50,51].

#### 2.5.1. Neutral GSL

Full scan mass spectra in the positive ion ESI mode were acquired for all peaks corresponding to investigate neutral GSL subclasses. Precursor ions of neutral GSL are commonly singly charged protonated molecules, thus MS/MS spectra of [M+H]^+^ ions were recorded and interpreted. The elucidation of MS/MS spectra of representative neutral GSL species within each investigated subclasses is described in Figure 5, as well as description of MS^3^ spectra of illustrative Gb_3_ and Gb_4_ species is shown in Appendix A. In addition, MS/MS spectra of other lipid species are illustrated in the following figures: HexCer (Appendix A), Hex_2_Cer (Appendix A), Gb_3_ (Appendix A), and Gb_4_ (Appendix A).

MS/MS spectra of HexCer, 18:1/16:0, Hex_2_Cer, 18:1/16:0, Gb_3_, 18:1/16:0, and Gb_4_, 18:1/16:0 were characterized by the fragmentation of precursor ions [M+H]^+^ identified as *m/z* 700.6 for HexCer (Figure 5A), *m/z* 862.6 for Hex_2_Cer (Figure 5B), *m/z* 1024.7 for Gb_3_ (Figure 5C), and *m/z* 1227.8 for Gb_4_ (Figure 5D). All observed fragment ions are singly charged.

The low abundant product ions Y_3_ at *m/z* 1024.7 (loss of N-acetylhexosamine, HexNAc), Y_2_ at *m/z* 862.6 (loss of hexose, Hex), Y_1_ at *m/z* 700.6 (loss of Hex), and Y_0_ at *m/z* 538.5 (loss of Hex) well describe a sequential cleavage of individual monosaccharide residues attached to the ceramide backbone of GSL molecule. Similarly, the Y_0_ ion is observed for HexCer, Y_0_–Y_1_ ions for Hex_2_Cer, Y_0_–Y_2_ ions for Gb_3_, and Y_0_–Y_3_ ions for Gb_4_. Y-type ions are accompanied by more abundant Z-type ions that are consistent with the additional loss of H_2_O from the corresponding Y-type ion (i.e., Δ*m/z* = 18). The observed Z_0_ ions at *m/z* 520.5 are derived from the cleavage of all saccharide units from GSL molecule, which corresponds to the structure of ceramide backbone.

MS/MS spectra also contain specific N-type fragment ion at *m/z* 264.3 confirming the presence of 18:1 sphingosine residue (denoted as N^II^). The *m/z* values of N^II^ fragment ions for common sphingosine bases are summarized in Table 1.

Although MS/MS spectra of Gb_3_ and Gb_4_ (Figure 5C,D) do not include the N-type fragment ion due to the low mass cut-off, the presence of this ion was further confirmed by performing MS^3^ experiment, where the ion at *m/z* 520.5 was fragmented for both Gb_3_ and Gb_4_ (Appendix A). The origin of N-type fragment ion (i.e., N^II^), along with other low abundant fragment ions (e.g., S, T, and U) sporadically observed in MS/MS spectra, is illustrated in Figure 6.

Additionally, the fragmentation of Gb_4_ resulted in one B-type fragment ion, i.e., B_2_ ion at *m/z* 366.1 (Figure 5D), confirming the cleavage of 4-linked HexNAc-Hex from Hex-Hex-Cer residue. The sequential cleavage of monosaccharide residues results in B, Y, and Z ions, which correlate well with the identification of the saccharide chain, therefore the fragmentation behavior of neutral GSL is predictable and provides explicit structural information about the saccharide sequence in the hydrophilic part of GSL molecules. The list of characteristic fragment ions that can be observed in positive ion ESI-MS/MS spectra for selected neutral GSL species is shown in Appendix A. The MS/MS fragmentation pattern of neutral GSL can be used as a guide for sequencing of the saccharide part.

#### 2.5.2. Acidic GSL

Full scan mass spectra in the negative ion ESI mode are acquired for SHexCer, SHex_2_Cer and GM_3_ belonging to the investigated group of acidic GSL. Precursor ions of acidic GSL are commonly singly charged deprotonated molecules, thus MS/MS spectra of [M-H]^−^ ions are measured. The interpretation of MS/MS spectra of illustrative GM_3_ and SHexCer species are depicted in Figure 7A,B, respectively. MS/MS spectra of other GM_3_ and SHexCer species detected in human plasma are illustrated in Appendix A, respectively. MS/MS spectra of SHex_2_Cer are not shown due to very poor fragmentation even when higher collision energy was applied. Obtained MS/MS spectra were characterized by only one specific fragment ion at *m/z* 403.1 (i.e., SulfoGal-Glc residue) with a very low abundance, which, however, was not observed in some lipid species. Therefore, we were not able to clearly describe the structures of SHex_2_Cer, although this GSL subclass is evidently present in human plasma.

The MS/MS spectrum of the most abundant ganglioside GM_3_ 34:1 was characterized by the fragmentation of precursor ion [M-H]^−^ at *m/z* 1151.7 (Figure 7A). Owing to the presence of only one sialic acid in GM_3_ molecule, all observed fragment ions are singly charged. The most abundant product ion Y_2_ is observed at *m/z* 860.5 and represents the cleavage of sialic acid from [M-H]^−^ ion. This ion is accompanied by abundant Y_1_ ion at *m/z* 698.5 (loss of terminal Hex) and Y_0_ ion at *m/z* 536.5 (loss of remaining Hex), which clearly characterizes the saccharide core of GM_3_. Z- and Y-type ions are generated in the same manner as described for neutral GSL, however, in case of GM_3_, the Z-type ions are commonly low abundant or may be absent, as shown in Figure 7A. On the other hand, the [M-H-H_2_O]^−^ ion at *m/z* 1133.7 derived from the probable loss of hydroxyl group as H_2_O from distinct sites on the GM_3_ molecule and Z_1_ ion at *m/z* 680.5 resulting from the loss of NeuAcHex are observed as well. These types of ions are also invariably present in all detected GM_3_ species. The presented MS/MS spectrum is also characterized by the typical [M-H-CO_2_]^−^ ion at *m/z* 1107.7, which corresponds to the neutral loss of carbon dioxide from the precursor ion. The abundant ion at *m/z* 930.5 denoted as ^0,2^X_2_ is the cross-ring cleavage product within sialic acid. Furthermore, the fragmentation of precursor ion conducted to the very low abundant B_3_ ion at *m/z* 614.2 (i.e., loss of H_2_O from the C_3_ ion) and C_3_ ion at *m/z* 632.2 (i.e., loss of NeuAcHexHex residue). Herein presented fragmentation pattern of GM_3_ species is in accordance with results published in previous works [45,52,53,54], and hence it confirms our findings.

The MS/MS spectrum of the most abundant sulfatide SHexCer 18:1/16:0 OH was specified by the fragmentation of the precursor ion [M-H]^−^ at *m/z* 794.5 (Figure 7B), and all observed product ions were singly charged. MS/MS spectra of sulfatide species with the sphingosine base 18:1 alongside an α-hydroxy-fatty acid substituent are typical of the ion cluster formation (*m/z* 522, 540 and 568). The ion cluster is not observed at comparable abundance in the same spectra of non-hydroxylated fatty acid-containing sulfatide species, as previously reported in works of Hsu and Turk [55,56]. The non-hydroxylated sulfatide species are commonly characterized by low or medium abundant specific fragment ions corresponding to the loss of fatty acid from precursor ion [M-H-FA]^−^ and additional loss of water [M-H-FA-H_2_O]^−^. The distinction between MS/MS spectra of hydroxylated and non-hydroxylated sulfatide species is also well demonstrated in the previous work [57]. Moreover, shifts of *m/z* of ion cluster may be observed, if the diverse sphingosine bases are present, as shown in Table 2. The MS/MS spectrum illustrated in Figure 7B demonstrates the formation of above mentioned ion cluster confirming the presence of an α-hydroxylated fatty acid substituent and the sphingosine base 18:1. The most abundant ion at *m/z* 568.2, which resulted from the cleavage of fatty acyl moiety as an aldehyde from [M-H]^−^, undergoes the loss of H_2_O yielding the low abundant ion at *m/z* 550. The ion at *m/z* 540.2 arises from the direct loss of fatty acyl as ketene from the [M-H]^−^ ion and continue through the dehydration process as well, yielding the ion at *m/z* 522. The ceramide part of the sulfatide is further elucidated by the presence of low abundant Y_0_ ion at *m/z* 552.5 and Z_0_ ion at *m/z* 534.5, which corresponds to the ceramide type 34:1 OH. Furthermore, the observed T-ion at *m/z* 296.2 and S-ion at *m/z* 312.2 clarify the occurrence of N-linked hydroxylated fatty acyl 16:0 OH. Lastly, the C_1_ ion at *m/z* 259.0 and B_1_ ion at *m/z* 241.0 correlate with the SHex residue lost from the ceramide backbone.

#### 2.5.3. Phosphatidylinositols and Lysophosphatidylinositols

The full scan mass spectra of phospholipids in the negative ion ESI mode are recorded for PI and LPI together with their MS/MS spectra of singly charged [M-H]^−^ precursor ions that show an extensive fragmentation, as demonstrated in the following figures: PI (Appendix A) and LPI (Appendix A). The clarification of MS/MS spectra is based on the structure-specific fragment ions described below.

MS/MS spectra of PI are characterized by the low abundant fragment ion corresponding to the loss of inositol (In) from [M-H]^−^ ion (denoted as [M-H-In]^−^) and two most abundant product ions obtained by the neutral loss of R_2_COOH group (denoted as PI-2B) and the neutral loss of R_2_COOH group with inositol (denoted as PI-2D) from [M-H]^−^ ion. These fragment ions are accompanied by low abundant ions resulting from the neutral loss of R_1_COOH group (denoted as PI-1B) and the neutral loss of R_1_COOH group with inositol (denoted as PI-1D) from precursor ion. A significant difference in the ratio of intensities between [R_1_COO]^−^ (denoted as *sn*-1) and [R_2_COO]^−^ (denoted as *sn*-2) ions is evident and indicates that the [R_1_COO]^−^ ion is more abundant and frequently one of the most abundant ions in MS/MS spectra. The preferential detection of product ions denoted as PI-2B and PI-2D as well as *sn*-1 positioned fatty acyl anion has been previously reported by Rovillos et al. [58] and is in agreement with our identification. Fragment ions resulting from the loss of fatty acyl chains as ketenes R_1_CH=C=O (denoted as PI-1A) and R_2_CH=C=O (denoted as PI-2A), that go hand in hand with fragment ions obtained by the loss of fatty acyl chains as ketenes R_1_CH=C=O (denoted as PI-1C) and R_2_CH=C=O (denoted as PI-2C) alongside inositol from [M-H]^−^ ion, were observed as well. The ions obtained by the loss of CO_2_ from R_2_COO^−^ ion, which resulted in polyunsaturated fatty acids (PUFA), were occasionally observed as well.

Similarly, the product ions for LPI are analogous to product ions of PI, aside from fragments that are not present due to absence of *sn*-1 or *sn*-2 positioned fatty acyl moiety at 2-LPI regioisomers, respectively 1-LPI regioisomers. In comparison with MS/MS spectra of PI, MS/MS spectra of LPI exhibit the [R_1_COO]^−^ ion as the most abundant product ion for the most LPI species alongside the less abundant ion denoted as [M-H-In-H_2_O]^−^ derived from the loss of inositol and H_2_O from [M-H]^−^ ion. The associated ion denoted as [M-H-In]^−^ is also present in low abundance. In addition, the fragments derived from the loss of CO_2_ from R_1_COO^−^ ion (denoted as PUFA) are observed in some cases. The fragment ions denoted as PI-1C and PI-1D are absent or not observed due to the low mass cut-off. Fragments of phosphoinositol-H_2_O at *m/z* 315.05 (denoted as PI-H_2_O) and inositolphosphate ion-H_2_O at *m/z* 241.01 (denoted as IP-H_2_O) are observed in both PI and LPI. All other product ions, including phosphoinositol-2H_2_O at *m/z* 297.04 (denoted as PI-2H_2_O), inositolphosphate ion at *m/z* 259.02 (denoted as IP) and inositolphosphate ion-2H_2_O at *m/z* 223.00 (denoted as IP-2H_2_O), are low abundant or absent.

In principle, characteristic fragments interpreted in MS/MS spectra, which depict the fragmentation pathways of particular lipid subclasses, correspond to the logical losses and follow regularities that apply to the fragmentation of individual parts of the molecule, and therefore supports the applicability of the method presented herein for the qualitative analysis of neutral and acidic GSL together with PI and LPI in one run.

### 2.6. Profile of Lipid Species in Human Plasma

Having characterized almost all lipid species within investigated lipid subclasses that have been detected in plasma extracts, we generated the profile of lipids in human plasma (Figure 8). The examination of MS and MS/MS spectra has revealed that HexCer 18:1/16:0 (34:1), Hex_2_Cer 18:1/16:0 (34:1), Gb_3_ 18:1/16:0 (34:1), Gb_4_ 18:1/16:0 (34:1), SHexCer 18:1/16:0 OH (34:1 OH), SHex_2_Cer 34:1, GM_3_ 34:1, PI 18:0/20:4 (38:4), PE 38:4 1-LPI 18:0, 2-LPI 20:4, 1-LPE 18:2, and 2-LPE 20:4 are the most abundant species within particular lipid subclass alongside numerous other lipid species that are slightly or considerably lower abundant, as shown in Figure 8. It is also evident (see Appendix A) that the majority of lipid species identified in human plasma have the 18:1 sphingosine base alongside the less abundant 16:0, 16:1, 17:0, 18:0, 18:2, and 18:0 OH sphingosine bases.

The lipid species reported without MS/MS confirmation due to sensitivity issues are labeled by an asterisk (Figure 8), which provides a clear evidence of the confidence of individual identifications. Abundances of individual lipid species in the lipid profile of plasma are shown as a function of sum composition. Intensities are shown as mean ± standard deviation based on 3 replicates.

Overall, we have identified and structurally characterized 59 neutral (+14 species not confirmed by MS/MS) and 64 acidic (+17 species not confirmed by MS/MS) GSL species together with 77 phospholipid species in human plasma (Figure 9). 37 of all 77 phospholipid species identified in human plasma were not confirmed by MS/MS, as phospholipids PE and LPE (35 lipid species in total) were not the principal subject of this study and do not interfere with any important glycosphingolipid subclass. However, our finding are consistent with PE and LPE species reported by Bang et al. [18] and Quehenberger et al. [59]. The complete list of individual lipid species identified in human plasma is shown in Appendix A. The fatty acyl level is stated, wherever it was possible to confirm it by MS/MS experiment. We cannot ruled out the possible presence of positional isomers and branching.

### 2.7. Extraction Recovery

Due to the intended future use and to prove that the methodology presented in this qualitative study can be further validated and used for the quantitative study of GSL in real samples without any or minor limits, extraction recovery experiments based on the signal responses of available IS for included lipid subclasses on medium concentration level were performed as well.

The extraction recovery was assessed by the comparison of absolute intensities of ISs in plasma samples spiked with 25 µL of IS Mix before and after the extraction in triplicates. We used one or two internal standard for each lipid subclass as shown in Table 3. The results expressed as mean ± RSD (Relative Standard Deviation) are illustrated in Figure 10. Evaluating the outcomes, we conclude that our extraction method is sufficiently efficient in extraction of GSL, which is proven by reproducible responses for all deuterated IS for particular GSL subclasses and ceramides (i.e., RSD from 4.0 to 12.5%). In addition, most of non-deuterated IS of GSL used as second IS also met the requirement (i.e., RSD ≤ 15%) [60,61], except for GlcCer (RSD 18.8%), GM_3_ (RSD 17.0%), and Cer (RSD 15.8%), where the RSD values were slightly higher. On the other hand, the extraction of phospholipids is slightly less effective than for GSL with a greater variability in the reproducibility (i.e., RSD from 7.5 to 21.8%), notably LPE subclass (RSD 47.8%).

We also performed the analysis of plasma using only protein precipitation, i.e., without SPE extraction. The protein precipitation was performed only in one replicate since we assumed signal suppression. By analyzing the sample, we found out there was significant signal suppression (up to 7 times lower response) for almost all examined lipid subclasses, excluding ceramides. We attributed the suppression effect to interfering compounds originating from the sample, especially salts. As a result, we conclude that SPE extraction should be unconditionally required in the sample preparation for GSL, because it removes salts that are likely to cause the signal suppression and therefore considerably improves the sensitivity.

## 3. Materials and Methods

### 3.1. Chemicals and Standards

Acetonitrile, acetone (both LC/MS grade), hydrochloric acid (purity for trace analysis, 36%), ethanol (gradient grade for LC), and glacial acetic acid (eluent additive for LC/MS) were purchased from Merck KGaA (Darmstadt, Germany). Methanol (LC/MS grade) and ammonium acetate (trace select) were purchased from Honeywell (Charlotte, NC, USA). Deionized water was prepared with a Barnstead Smart2Pure Water Purification System (Waltham, MA, USA).

The TLC neutral GSL mixture (bovine and porcine origin) was purchased from Matreya LLC (State College, PA, USA). The internal standards (IS) for each investigated lipid subclass were purchased commercially (Avanti Polar Lipids, Alabaster, AL, USA; Matreya LLC, Pennsylvania, PA, USA; Merck, Darmstadt, Germany) or were prepared by the synthesis (see reference [46]). Internal standards mixture (IS Mix) was prepared by mixing selected deuterated and non-deuterated IS and diluting them with methanol to obtain final IS concentrations shown in Table 3.

### 3.2. Collection and Processing of Blood Samples

Fresh blood samples (9 mL) were drawn from healthy male donors with the age between 20 and 50. Blood was collected into 9 mL K_3_-EDTA Vacuette tubes (Dialab, Czech Republic) and plasma was separated from other blood components (i.e., platelets, erythrocytes and leukocytes) by centrifugation of blood samples at 2500× *g* for 10 min under refrigerated conditions (i.e., 4 °C) immediately after blood collection. Supernatants were then collected to obtain homogeneous plasma sample. Subsequently, aliquots of 1.5 mL plasma sample were transferred into 1.8 mL Eppendorf tubes, immediately frozen and stored at −80 °C prior to further sample processing and analysis. The study was conducted in accordance with the Declaration of Helsinki, and the protocol was approved by the Ethics Committee at University Hospital Olomouc, Czech Republic (reference number 57/15). All subjects gave their informed consent for inclusion before they participated in the study.

### 3.3. Sample Preparation

The sample of 250 µL of human plasma was homogenized and deproteinized in 3 mL of ethanol using ultrasonic bath at 40 °C for 10 min. Then, 600 µL of deionized water was added, the mixture was vortexed for 1 min and centrifuged at 10,000 rpm for 3 min under ambient conditions. The supernatant containing lipids was collected, evaporated in a heated block at 35 °C with a gentle stream of nitrogen to dryness and redissolved in 1 mL of deionized water. Deproteinized and redissolved plasma was purified and concentrated using SPE, where seven C18-based columns (i.e., DSC-18 (A), DSC-18 (B), DSC-18Lt, SepPak-tC18. ENVI-18, Strata C18-E, Spe-ed C18/18), one C8-based column (i.e., DSC-8), three polymeric columns (i.e., SDB-L, Strata X, Oasis HLB), one normal phase column (i.e., Diol) and two ZrO_2_-based colums (i.e., Phree, HybridSPE-phospholipid) from several manufacturers were tested using protocols recommended by vendors.

For the assessment of extraction recovery, the IS Mix was added to plasma in the volume of 25 µL (i.e., medium concentration level) before and after the extraction procedure. The volume used for spiking was chosen so that the resulting concentration of IS in plasma approximately corresponds to concentrations of medium abundant lipid species identified in plasma within a particular lipid subclass. The extraction recovery expressed as percentage was calculated from the ratio of the signal intensity of IS in the sample spiked after the extraction to the signal intensity of IS in the sample spike before the extraction and multiplied by 100.

First, 500 mg of Spe-ed C18/18 cartridge with 40 µm particle size and 60 Å porosity (Applied Separations, Allentown, PA, USA) was conditioned with 3 × 1 mL of methanol and equilibrated with 3 × 1 mL of deionized water. Then, 1 mL of sample dissolved in deionized water was loaded onto the cartridge and washed with 3 × 1 mL of deionized water. Finally, the retained lipids were eluted by 3 × 1 mL of methanol. The eluate was collected, and then evaporated in a heated block at 35 °C with a gentle stream of nitrogen to dryness and redissolved in 50 µL of methanol to obtain purified lipid extract for the HILIC-ESI/MS analysis.

### 3.4. HPLC Conditions

HILIC separation of particular purified lipid extracts was performed on an Ascentis Si column (150 × 2.1 mm, 3 µm particle size; Sigma-Aldrich, St Louis, MO, USA) using a Dionex UltiMate 3000 series HPLC system equipped with thermostated WPS-3000 RS autosampler cooled to 7 °C (Thermo Scientific, Waltham, MA, USA). The temperature in a column chamber was set at 40 °C and controlled by a thermostat. The lipid extract was injected onto a chromatographic column and consequently separated using the binary gradient elution. The mobile phase flow rate was 0.5 mL/min, and the injection volume was 3 μL. The gradient elution program was set as follows: 0 min—10% of mobile phase B, 7 min—11.4% of B, 16 min—20% of B, 18 min—20% of B, where the mobile phase A was acetonitrile, and the mobile phase B was 10 mM ammonium acetate in 10% acetonitrile. Both mobile phases were slightly acidified by the addition of 0.5 µL of glacial acetic acid per 100 mL. The total run time including the equilibration was 25 min.

### 3.5. MS Conditions

The majority of experiments were accomplished on a Velos Pro series dual-pressure linear ion trap mass spectrometer equipped with an Ion Max source and HESI-II probe (Thermo Fisher Scientific, San Jose, CA, USA). The ESI settings were as follows: heater temperature 300 °C, spray voltage 3.0 kV, capillary temperature 300 °C, sheath gas (N_2_) flow rate 35 arbitrary units, and auxiliary gas (N_2_) flow rate 10 arbitrary units. The total nitrogen flow rate corresponded to 8 L/min. The MS spectra were acquired by scanning of the mass range *m/z* 50–1500 in both polarity modes, followed by dual-stage MS/MS experiments using the MS/MS mass list, where *m/z* values of individual lipid species were defined within scanning windows corresponding to the retention time of particular lipid subclass. The following setup of ion optics system and ion trap was used for MS and MS/MS analysis: S-lens voltage 60 V, F-lens voltage −9.0 V, split gate lens voltage −90 V, multipole MP00 rf lens offset −2 V, lens L0 voltage −3 V, multipole MP0 offset −9 V, lens L1 voltage −15 V, multipole MP1 offset −20 V, multipole RF amplitude 700 Vpp, normalized collision energy 40%, isolation width *m/z* 1.0, scan rate 33.3 Da/s, automatic gain control (AGC) with a maximum injection time 250 ms, activation Q 0.250, and activation time 10.0 ms. Selected MS/MS scans were followed by triple-stage tandem mass spectrometry (MS^3^), where the required peak of the MS/MS spectrum was further subjected to fragmentation under conditions described above. Thermo Scientific Xcalibur 4.2 and LTQ Tune Plus 2.8 software programs were used for data processing and instrument control. The ion trap was calibrated using Thermo Scientific Pierce LTQ Velos ESI positive ion and negative ion calibration solutions resulted in FWHM of 0.5 (full width at half maximum) within the scanning range and the mass accuracy of ±0.1 Da.

### 3.6. Data Analysis

The data file of mass spectra were acquired in profile mode (*.raw file) from Velos Pro mass spectrometer, in both positive and negative ion modes. First, MS spectra were obtained, and the scan range corresponding to the retention time window for each lipid subclass was subsequently determined by comparing the first and last eluting lipid species within a particular lipid subclass. Then, the defined scan range was applied for each lipid subclass in the sample, resulting in mass spectra with identification of all *m/z*. The obtained mass spectra were further averaged and modified using subtracting the background noise. Information of all detected peaks on averaged spectrum containing *m/z* values and signal intensities (peak heights) was exported to Excel file for further data analysis. Then tables of signal intensities corresponding to the particular *m/z* were filtered by comparing individual *m/z* values with our internal database for each lipid subclass and applying predefined mass tolerance window (0.1 *m/z*). Finally, the resulting filtered tables for each lipid subclass represent the natural lipid species in human plasma.

Lipid species identified within MS analysis were further subjected to MS/MS analysis. The MS/MS analysis was performed using the mass list tool, where the *m/z* values of identified lipid species together with the normalized collision energy were embedded into the corresponding scan range window. The presence and composition of individual lipid species were then confirmed by an in-depth examination of relevant MS/MS spectra.

## 4. Conclusions

In this study, the alternative lipid extraction method is applied for the isolation and purification of GSL from human plasma. This method eliminates the excessive use of harmful chemicals, minimizes analyte losses, and increases the detection sensitivity due to the use of higher volumes of plasma and the ability to extract GSL with a high yield. The obtained extraction recovery values further confirm the applicability of the modified monophasic solvent system for the isolation of GSL including less polar GSL classes. The work reports the identification of 123 GSL species in the human plasma, including those not yet previously reported. The structural assignment of most GSL species is elucidated by the interpretation of MS and MS/MS spectra including the detection of some highly polar phospholipid classes, which were not removed by the sample preparation steps (PI, LPI, PE, and LPE). Several previous works have shown the association between alterations of lipid species in plasma and the onset of some human diseases, thus the large attention has been devoted to the lipidomic characterization. However, GSL (especially Gb_3_ and Gb_4_) are mostly not covered in generic lipidomic methods, therefore our approach may serve for the extension of the GSL coverage in biological samples. The present work is focused on the qualitative analysis and lipid profiling, so the next step will be the quantitation of GSL using the suitable deuterated and exogenous IS for individual lipid subclasses including the full method validation similarly as for our previous works developed for high-throughput lipidomic screening [46,62,63].

## Figures and Tables

**Figure 1 metabolites-11-00140-f001:**
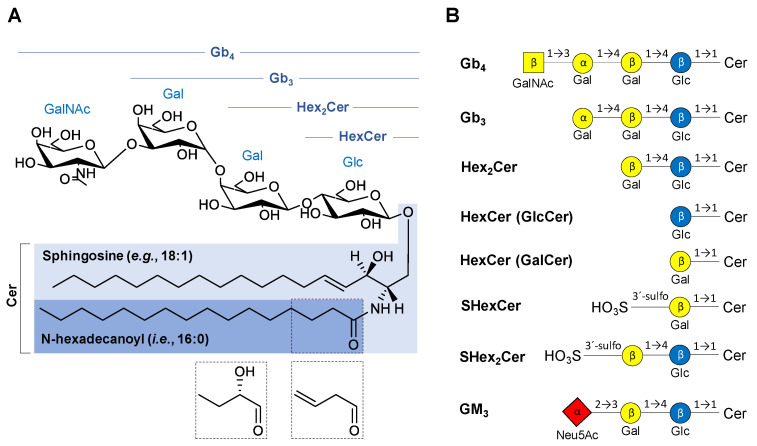
Overview of chemical structures of the neutral and acidic glycosphingolipid (GSL) subclasses studied in this work: (**A**) General structure of neutral GSL depicting individual parts of the complex molecule with possible modifications of N-linked fatty acyl chain (dashed boxes). (**B**) Basic structures of GSL subclasses using the established glycan symbols (the letter and the number within symbols define the nature of the glycosidic linkage). Abbreviations used in the figure are as follows: ceramides (Cer), N-acetylneuraminic acid (Neu5Ac), galactose (Gal), glucose (Glc), N-acetylgalactosamine (GalNAc), globotetraosylceramides (Gb_4_), globotriaosylceramides (Gb_3_), dihexosylceramide (Hex_2_Cer), monohexosylceramide (HexCer)—glucosylceramide (GlcCer) or galactosylceramide (GalCer), monohexosylsulfatide (SHexCer), dihexosylsulfatide (SHex_2_Cer), monosialodihexosylganglioside (GM_3_).

**Figure 2 metabolites-11-00140-f002:**
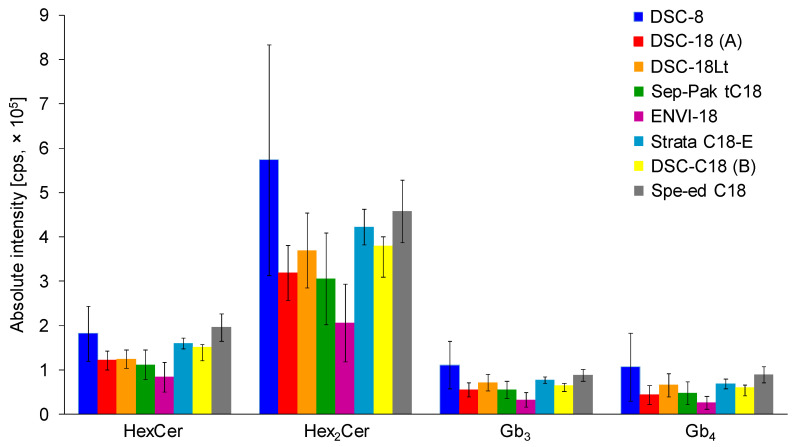
Comparison of C18-based solid-phase extraction (SPE) columns from different vendors using the same extraction protocol (see details in Experimental section). Bar graphs display means of absolute peak intensities of the most abundant species within particular GSL subclasses, and error bars illustrate standard deviations for 9 analyses (each SPE extraction was performed in triplicate and 3 measurements were performed for each sample).

**Figure 3 metabolites-11-00140-f003:**
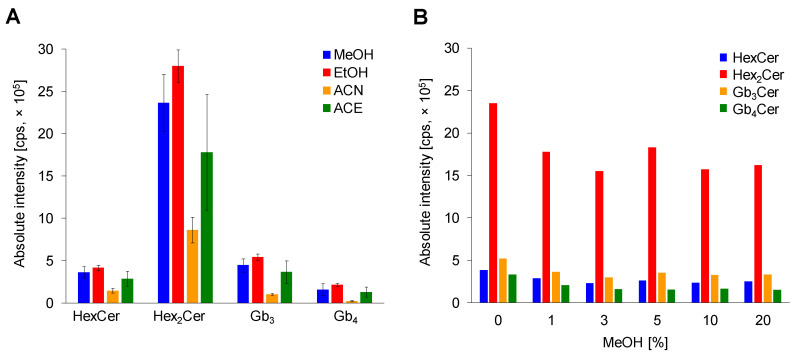
Optimization of sample preparation. (**A**) Comparison of solvents methanol (MeOH), ethanol (EtOH), acetonitrile (ACN) and acetone (ACE) for plasma deproteinization. Bar graphs display means of absolute peak intensities of the most abundant species within particular GSL subclass, and error bars illustrate standard deviations within nine analyses. (**B**) Effect of MeOH % abundance in loading step.

**Figure 4 metabolites-11-00140-f004:**
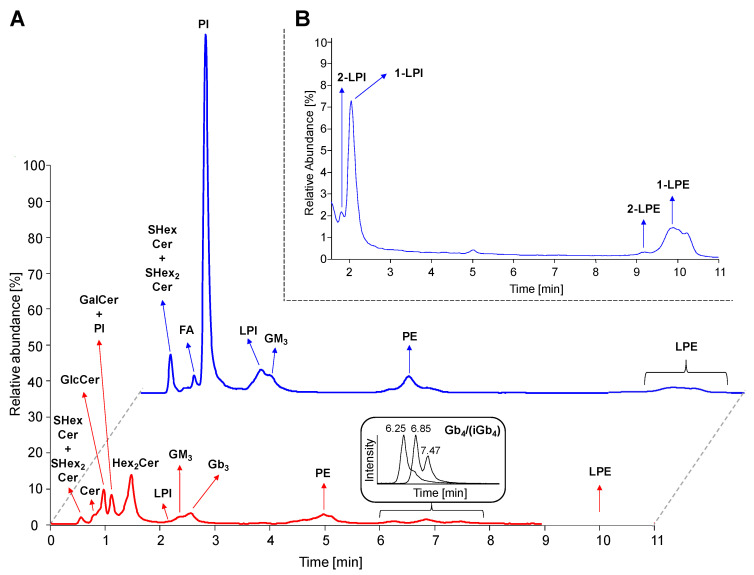
Hydrophilic interaction liquid chromatography coupled to electrospray ionization mass spectrometry (HILIC-ESI/MS) chromatograms. (**A**) Reconstructed ion current (RIC) chromatograms of detected lipid subclasses (*m/z* 50–1500) in human plasma using both positive (bottom red line) and negative (upper blue line) polarity modes. (**B**) Chromatogram illustrating separation of *sn*-1 (i.e., 1-LPI and 1-LPE) and *sn*-2 (i.e., 2-LPI and 2-LPE) regioisomers of lysophospholipids. Abbreviations: (lyso)phosphatidylinositols, (L)PI; (lyso)phosphatidylethanolamines, (L)PE; isoglobotetraosylceramides (iGb_4_), and fatty acids (FA).

**Figure 5 metabolites-11-00140-f005:**
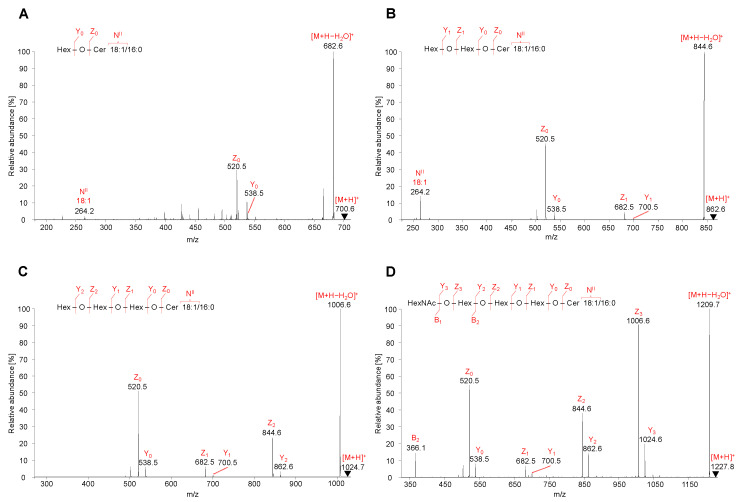
HILIC-ESI-MS/MS spectra of representative neutral GSL species. Tandem mass spectrometry (MS/MS) spectra illustrate fragmentation pathways with specific product ions of (**A**) HexCer 18:1/16:0 at *m/z* 700.6, (**B**) Hex_2_Cer 18:1/16:0 at *m/z* 862.6, (**C**) Gb_3_ 18:1/16:0 at *m/z* 1024.7, and (**D**) Gb_4_ 18:1/16:0 at *m/z* 1227.8.

**Figure 6 metabolites-11-00140-f006:**
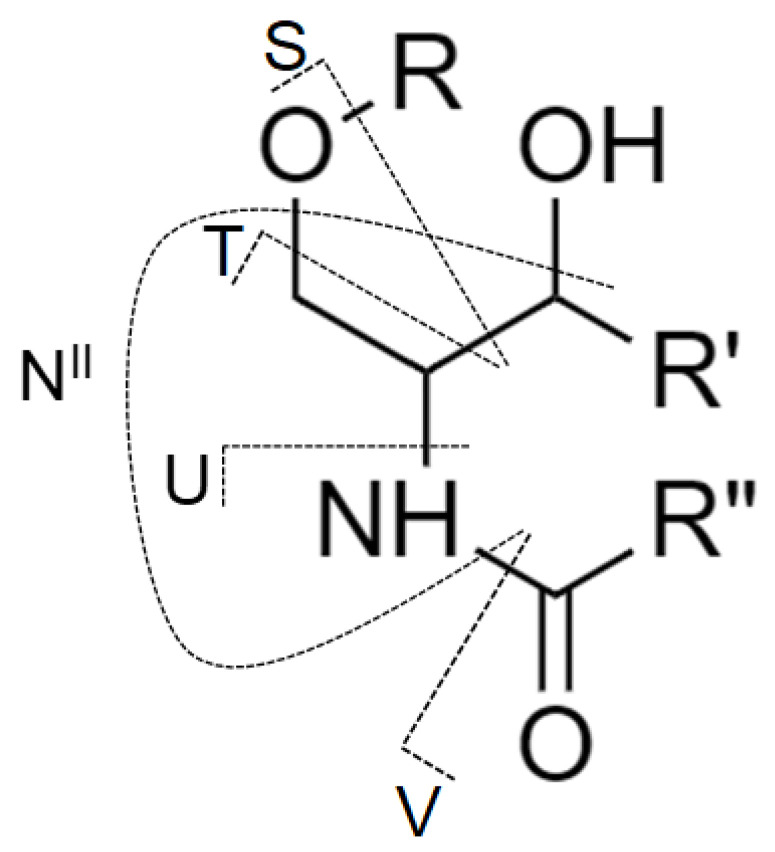
The nomenclature for cleavages of the ceramide part of glycosphingolipid molecule used in this work and originally proposed by Ann and Adams [51] with the addition from Merrill et al. [9].

**Figure 7 metabolites-11-00140-f007:**
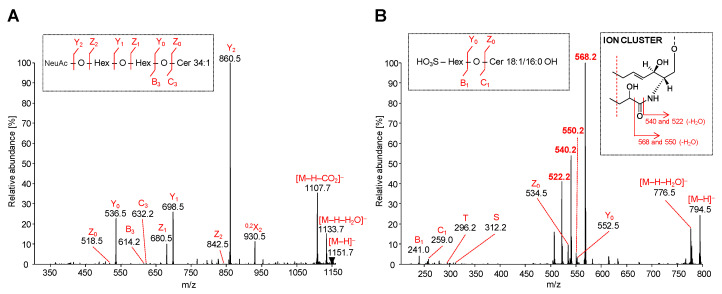
HILIC-ESI-MS/MS spectra of representative acidic GSL species: MS/MS spectra illustrate fragmentation pathways with specific product ions of (**A**) ganglioside GM_3_ 34:1 at *m/z* 1151.7, and (**B**) sulfatide SHexCer 18:1/16:0 OH at *m/z* 794.5.

**Figure 8 metabolites-11-00140-f008:**
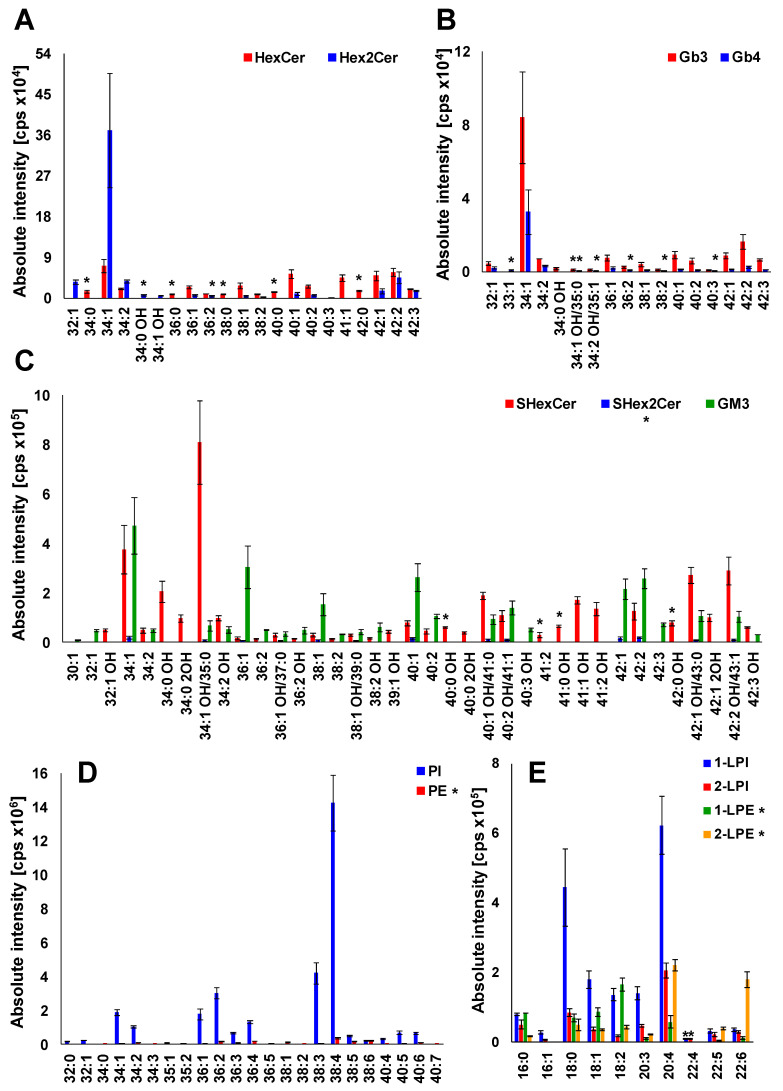
Profile of human plasma lipids. (**A**) mono- and dihexosylceramides, (**B**) globotriaosyl- and globotetraosylceramides, (C) sulfatides and gangliosides, (**D**) phospholipids, and (**E**) lysophospholipids. The individual lipid species within particular lipid subclass were identified and structurally characterized by the interpretation of MS/MS spectra (see Figure 5, Figure 6 and Appendix A). Bar graphs represent the mean intensities (based on 3 replicates) as a function of sum composition with error bars corresponding to relative standard deviation (RSD). Lipid species labeled by an asterisk were not confirmed by MS/MS experiments.

**Figure 9 metabolites-11-00140-f009:**
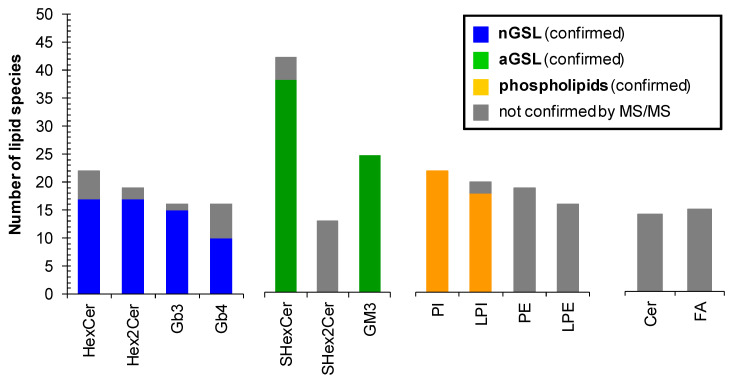
Number of lipid species identified within particular lipid subclass. Colored bars represent the number of lipid species identified and structurally characterized by MS/MS experiments. Grey bars correspond to identified lipid species that were not confirmed by MS/MS experiments. Abbreviations: neutral glycosphingolipids (nGSL) and acidic glycosphingolipids (aGSL).

**Figure 10 metabolites-11-00140-f010:**
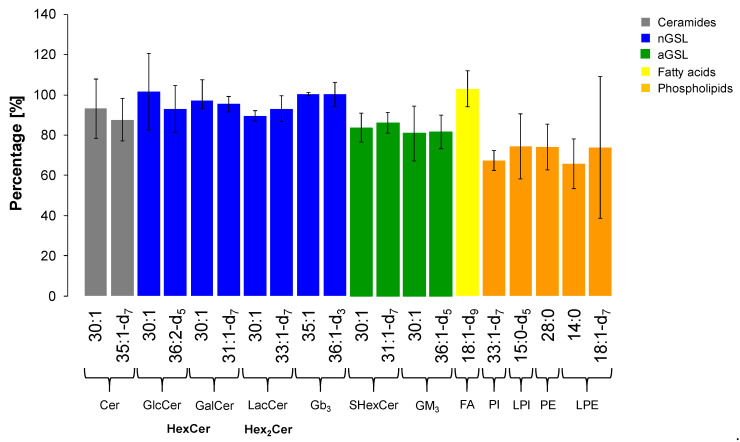
Extraction recovery for each internal standard investigated within particular lipid subclass. The values are expressed as mean ± RSD (*n* = 3).

**Table 1 metabolites-11-00140-t001:** Common sphingosine bases and their *m/z* values for N^II^ fragment in positive ion mode.

**Sphingosine base**	14:1	16:1	18:0	18:1	18:2	20:1	18:0 OH
***m/z* of N^II^ ion**	208.2	236.2	266.3	264.3	262.3	292.3	282.3

**Table 2 metabolites-11-00140-t002:** Characteristic product ions of hydroxylated sulfatides (i.e., ion cluster).

Sphingosine Base	Characteristic Product Ion [*m/z*]		
	Cleavage of Fatty Acid from Carbonyl Group Site	Additional Elimination of CO	Additional Elimination of H_2_O
14:1	512	484	466
16:0	542	514	496
16:1	540	512	494
18:0	570	542	524
18:1	568	540	522
18:2	566	538	520
20:0	598	570	552
20:1	596	568	550
18:0 OH	586	558	540

**Table 3 metabolites-11-00140-t003:** Internal standards (IS) concentrations in internal standard mixture (IS Mix) and plasma for individual lipid subclasses.

Lipid Class	Internal Standard (Ceramide Composition)		Concentration in IS Mix (μg/mL)	Concentration in Plasma (μg/mL)
Cer	18:1/12:0 18:1-d_7_/17:0	30:1 35:1-d_7_	0.6 1.5	0.06 0.15
GalCer (HexCer)	18:1/12:0 18:1-d_7_/13:0	30:1 31:1-d_7_	0.2 0.6	0.02 0.06
GlcCer (HexCer)	18:1/12:0 18:1-d_5_/18:1	30:1 36:2-d_5_	3 5	0.3 0.5
LacCer (Hex_2_Cer)	18:1/12:0 18:1-d_7_/15:0	30:1 33:1-d_7_	14 7	1.4 0.7
Gb_3_	18:1/17:0 18:1/18:0-d_3_	35:1 36:1-d_3_	3.5 4.5	0.35 0.45
Gb_4_	not yet commercially available			
GM_3_	18:1/12:0 18:1/18:0-d_5_	30:1 36:1-d_5_	10 12	1 1.2
SHexCer	18:1/12:0 18:1-d_7_/13:0	30:1 31:1-d_7_	0.45 0.45	0.045 0.045
SHex_2_Cer	not yet commercially available			
FA	18:1-d_9_	---------	150	15
PI	18:1-d_7_/15:0	33:1-d_7_	30	3
LPI	15:0-d_5_	---------	1.5	0.15
PE	14:0/14:0	28:0	3.75	0.375
LPE	14:0 18:1-d_7_	--------- ---------	3 2.5	0.30 0.25

## Data Availability

Untargeted lipidomic profiling and identification was performed at the Analytical Laboratory, University of South Bohemia in České Budějovice, Faculty of Science, Institute of Chemistry, České Budějovice, Czech Republic, where the raw data are available on request.

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
