# Peer review of "Comprehensive Identification of Glycosphingolipids in Human Plasma Using Hydrophilic Interaction Liquid Chromatography—Electrospray Ionization Mass Spectrometry"

_metabolites, 2021, doi:10.3390/metabo11030140_

Round 1

Reviewer 1 Report

The manuscript titled "Comprehensive Identification of Glycosphingolipids in Human Plasma using Hydrophilic Interaction Liquid Chromatography – Electrospray Ionization Mass Spectrometry" submitted to Metabolites, describes a very interesting and useful approach to identification Glycosphingolipids isolated from human plasma. For extraction authors proposed an optimized monophasic ethanol–water solvent system capable to recover a broad range of GSL species, and for instrumental analysis obtained extract, authors proposed the hydrophilic interaction liquid chromatography (HILIC) coupled to electrospray ionization linear ion trap tandem mass spectrometry (ESI‐LIT‐MS/MS).  The authors presented in the manuscript identification of 154 GSL species within 7 lipid subclasses and 77 phospholipids representing, according authors knowledge, the highest number of GSL species ever reported in the human plasma. Also according authors, the work reports the identification of 123 GSL species in the human plasma, including those not yet previously reported. The study seems to be properly designed and authors input a lot of effort to reach valuable results and proper conclusions.

Regarding structure of manuscript and content: Abstract and introduction is well written divided into subsections. Methods and Materials are very well organize. Results and Discussion are presented very clearly and describe all experiments very well. Tables and figures are presented correctly, RSD are calculated and presented as well.

Author Response

  • Reviewer #1

    The manuscript titled "Comprehensive Identification of Glycosphingolipids in Human Plasma using Hydrophilic Interaction Liquid Chromatography – Electrospray Ionization Mass Spectrometry" submitted to Metabolites, describes a very interesting and useful approach to identification Glycosphingolipids isolated from human plasma. For extraction authors proposed an optimized monophasic ethanol–water solvent system capable to recover a broad range of GSL species, and for instrumental analysis obtained extract, authors proposed the hydrophilic interaction liquid chromatography (HILIC) coupled to electrospray ionization linear ion trap tandem mass spectrometry (ESI‐LIT‐MS/MS).  The authors presented in the manuscript identification of 154 GSL species within 7 lipid subclasses and 77 phospholipids representing, according authors knowledge, the highest number of GSL species ever reported in the human plasma. Also according authors, the work reports the identification of 123 GSL species in the human plasma, including those not yet previously reported. The study seems to be properly designed and authors input a lot of effort to reach valuable results and proper conclusions.

    Regarding structure of manuscript and content: Abstract and introduction is well written divided into subsections. Methods and Materials are very well organize. Results and Discussion are presented very clearly and describe all experiments very well. Tables and figures are presented correctly, RSD are calculated and presented as well.

    • Thank you very much for positive comments.

Reviewer 2 Report

The manuscript by Hořejší et al. provide a systematic analysis to identify glycosphingolipids using HILIC chromatography and MS analysis. The study is very well designed that evaluates both pre-analytical and analytical factors needed to be taken into consideration for measurement of glycsphingolipids. I only have few minor comments that needs to be addressed prior to its publication.

  1. Data Analysis: Authors should include a section on data analysis in the methods. As the authors have used a low resolution instrument this section will be helpful for readers to follow the process employed by authors to unambiguously identify lipids in a complex sample such as plasma.
  2. Line 509 Figure “8” is missing
  3. Line 615-616 What do the authors mean by “volume of 25 ul as medium concentration level”? Why did the authors performed spiking of IS mix both before and after the extraction procedure to the sample and how did it allow them to calculate recovery? Further expand this section and provide formula used to calculate recovery.
  4. Table S2 Do the authors suggest that they have a precision of 4 decimal places for the observed fragment ions or did they just use default settings in instrument control software to generate this table.
  5. Table S3 Provide some form of abundance metric (absolute intensity, percentage of the most abundant analyte) for each of the analyte. While the bar plots (Figure 8 and 9) are good representation but most of the analytes are at the lower end of y-axis and therefore hard to evaluate abundance difference among themselves. Authors should also add a column on observed mass and mass error to the table in addition to the theoretical mass.

Author Response

Reviewer #2

The manuscript by Hořejší et al. provide a systematic analysis to identify glycosphingolipids using HILIC chromatography and MS analysis. The study is very well designed that evaluates both pre-analytical and analytical factors needed to be taken into consideration for measurement of glycosphingolipids. I only have few minor comments that needs to be addressed prior to its publication.

    Data Analysis: Authors should include a section on data analysis in the methods. As the authors have used a low resolution instrument this section will be helpful for readers to follow the process employed by authors to unambiguously identify lipids in a complex sample such as plasma.

  • The section “Data analysis” was added to the manuscript as suggested by this reviewer.

    Line 509 Figure “8” is missing

  • We have revised it.

    Line 615-616 What do the authors mean by “volume of 25 ul as medium concentration level”? Why did the authors performed spiking of IS mix both before and after the extraction procedure to the sample and how did it allow them to calculate recovery? Further expand this section and provide formula used to calculate recovery.

  • By “volume of 25 µl as medium concentration level”, it is meant that we used 25 µL of IS Mix containing such amount of IS, which approximately corresponded to the amount of medium abundant lipid species identified in plasma within particular lipid subclass.
  • We performed spiking of IS Mix before and after the extraction to calculate the extraction recovery of individual IS in plasma samples based on the comparison of signal intensities in sample spiked after the extraction to the sample spiked before the extraction.
  • We have expanded this section in the manuscript to answer the questions raised by this reviewer as well as to make this part easier to understand for readers.

    Table S2 Do the authors suggest that they have a precision of 4 decimal places for the observed fragment ions or did they just use default settings in instrument control software to generate this table.

  • No, since we used a low-resolution instrument, we have a precision of one decimal place. The presented table was generated using the Lipid maps tool “in-silico MS/MS peak prediction for glycosphingolipids” and illustrates the theoretical m/z of possible fragment ions that may be observed in MS/MS spectra of neutral GSL. Based on this question, we realized this may be confusing for readers. Therefore, we slightly modified the table title as follows “Characteristic fragment ions (theoretical m/z) of selected neutral GSL species with the most common sphingosine base 18:1 that may be observed in positive ion ESI-MS/MS spectra”, to avoid any misunderstanding.

    Table S3 Provide some form of abundance metric (absolute intensity, percentage of the most abundant analyte) for each of the analyte. While the bar plots (Figure 8 and 9) are good representation but most of the analytes are at the lower end of y-axis and therefore hard to evaluate abundance difference among themselves. Authors should also add a column on observed mass and mass error to the table in addition to the theoretical mass.

  • We have considered this initiative comment, added the abundance metric based on the mean absolute intensity and a column with observed masses (m/z). We did not provide mass error values as we used a low-resolution instrument having mass accuracy of one decimal place, as stated in Methods section. Therefore, the mass accuracy values would not be very beneficial in our case.

Reviewer 3 Report

The presented manuscript concerns the identification of glycosphingolipids in human plasma using HILIC assay coupled to ESI mass spectrometer. The introduction is well written, and it is setting the background regarding the GSL and their diagnostic potential. It is very thoroughly describing the current analytical lipidomics workflows. Although, scientifically it is very sound, stylistically is complicated and I would suggest shortening some sentences. It should help readers to understand better (long sentences examples: 'Both parts are immensely complex due to multiple variations in the monosaccharide composition, sequence, binding positions, branching [2; 11], modifications (e.g., sulfation or sialylation) [8] as well as the variation of hydrophobic part reflecting the type of ceramide base or N-acyl chain length and the degree of its unsaturation and hydroxylation [12].' or 'To make it further more complicated, variances in the lipid solubility must also be taken into account since some lipids are highly hydrophilic (e.g., more complex neutral GSL and gangliosides) and partitioning mostly to alcohol‐rich layer as they are so water‐soluble that they prefer aqueous phase of organic solvent extraction system [9; 31; 35], while others are less polar or hydrophobic (e.g., ceramides, less complex GSL, and sulfatides) and remain in non‐polar organic layer [9; 35].'). The Materials & Methods section describes all settings in very detail (even the ion optics settings inside the MS that is not so common).

Nevertheless, SPE vendors' affiliations are missing, which might complicate the future search of those SPE columns (Lines 610-614). The Results & Discussion section is very comprehensive and detailed, which I appreciate. The extent of Supplementary Data and overall presented results is admirable. On the other hand, I have to say that some of the information is too basic for the manuscript and thus redundant, e.g., lines 364-369 – 'The most abundant product ions [M+H‒H2O]+ in MS/MS spectra are generated by the loss of hydroxyl group as H2O from various sites of the GSL molecule. Although these product ions are non‐specific, they are commonly occurring as important peaks in tandem mass spectra of almost all oxygen containing functional groups in the positive ion mode. Moreover, the significance of these ions should [M+H−H2O]+ not be underrated, because their correct interpretation may be beneficial to the confirmation of determined molecular weight [52].'

In my opinion, it is just very long, and it might be difficult for the reader to grasp the main ideas, thus I would suggest reducing the size of the main manuscript. In the end, the overall manuscript is at the high analytical standard, and I have just a few questions and minor comments:

  • What was the exact criteria for 'Satisfactory' and 'Unsatisfactory' error bars as discussed in lines 222 – 224?
  • Have authors studied the in-source fragmentation and their relevance to identification and quantification of GSL?
  • Have authors observed any molecule rearrangements as a consequence of fragmentation in the spectra?
  • Have authors observed any radical-charged fragments within the MSMS or MS spectra?
  • The comparison with NIST SRM 1950 is, in my opinion, not very important since there was no quantification done. Usually, it can be expected a different set of lipid species based on the sample. My suggestion would be to remove this part, which should be beneficial in shortening the manuscript's length.
  • Line 509: Number at the figure description is missing.

Author Response

Reviewer #3

The presented manuscript concerns the identification of glycosphingolipids in human plasma using HILIC assay coupled to ESI mass spectrometer. The introduction is well written, and it is setting the background regarding the GSL and their diagnostic potential. It is very thoroughly describing the current analytical lipidomics workflows. Although, scientifically it is very sound, stylistically is complicated and I would suggest shortening some sentences. It should help readers to understand better (long sentences examples: 'Both parts are immensely complex due to multiple variations in the monosaccharide composition, sequence, binding positions, branching [2; 11], modifications (e.g., sulfation or sialylation) [8] as well as the variation of hydrophobic part reflecting the type of ceramide base or N-acyl chain length and the degree of its unsaturation and hydroxylation [12].' or 'To make it further more complicated, variances in the lipid solubility must also be taken into account since some lipids are highly hydrophilic (e.g., more complex neutral GSL and gangliosides) and partitioning mostly to alcohol‐rich layer as they are so water‐soluble that they prefer aqueous phase of organic solvent extraction system [9; 31; 35], while others are less polar or hydrophobic (e.g., ceramides, less complex GSL, and sulfatides) and remain in non‐polar organic layer [9; 35].').

  • We have shortened some long sentences to make it less complicated based on the recommendation of this reviewer.

The Materials & Methods section describes all settings in very detail (even the ion optics settings inside the MS that is not so common). Nevertheless, SPE vendors' affiliations are missing, which might complicate the future search of those SPE columns (Lines 610-614).

  • SPE vendors’ affiliations are stated in Table S1 in Supplementary material and the reference to this table is provided in the section 2.2.1.

The Results & Discussion section is very comprehensive and detailed, which I appreciate. The extent of Supplementary Data and overall presented results is admirable. On the other hand, I have to say that some of the information is too basic for the manuscript and thus redundant, e.g., lines 364-369 – 'The most abundant product ions [M+H‒H2O]+ in MS/MS spectra are generated by the loss of hydroxyl group as H2O from various sites of the GSL molecule. Although these product ions are non‐specific, they are commonly occurring as important peaks in tandem mass spectra of almost all oxygen containing functional groups in the positive ion mode. Moreover, the significance of these ions should [M+H−H2O]+ not be underrated, because their correct interpretation may be beneficial to the confirmation of determined molecular weight [52].'

  • We have removed this part from the manuscript based on the comment of this reviewer.

In my opinion, it is just very long, and it might be difficult for the reader to grasp the main ideas, thus I would suggest reducing the size of the main manuscript. In the end, the overall manuscript is at the high analytical standard, and I have just a few questions and minor comments:

  • We have considered this comment, and are aware of that the length of the manuscript is longer than usual as we put a lot of effort to describe the whole workflow in as much detail as possible even with regard to the amount of experimental work. Nevertheless, we have searched and removed some duplications and redundant information or reformulated some sentences to reduce the length of the manuscript at least a little.

    What was the exact criteria for 'Satisfactory' and 'Unsatisfactory' error bars as discussed in lines 222 – 224?

  • We used relative standard deviation (RSD ≤ 15 %) as a criterion for evaluation of variability illustrated by error bars. Although RSD for Spe-ed C18 cartridge was slightly higher compared to Strata C18-E and DSC-C18 (B) cartridges, we took into account the signal intensity as well and made a compromise between the highest possible intensity and acceptable RSD value.

    Have authors studied the in-source fragmentation and their relevance to identification and quantification of GSL?

  • We have not studied in-source fragmentation, but only the fragmentation in collision cell. We have not observed problems caused by in-source identification, but we may investigate this issue in more detail in our subsequent work on the quantitative aspects.

    Have authors observed any molecule rearrangements as a consequence of fragmentation in the spectra?

  • We cannot conclusively say, whether we observed any rearrangement of molecules in the spectra, as we focused almost exclusively on specific fragments defining the saccharide core and the main composition of the ceramide.

    Have authors observed any radical-charged fragments within the MSMS or MS spectra?

  • Since radical-charged fragments do not typically occur in atmospheric pressure ionization techniques and especially in the case of electrospray ionization are usually very rare, we did not take these fragments into specific focus. None of reported fragment ions is reported as radical ion.

    The comparison with NIST SRM 1950 is, in my opinion, not very important since there was no quantification done. Usually, it can be expected a different set of lipid species based on the sample. My suggestion would be to remove this part, which should be beneficial in shortening the manuscript's length.

  • We have removed this part based on the suggestion of this reviewer.

    Line 509: Number at the figure description is missing.

  • Number is added.